# Unveiling Key Genes and Crucial Pathways in Goose Muscle Satellite Cell Biology Through Integrated Transcriptomic and Metabolomic Analyses

**DOI:** 10.3390/ijms26083710

**Published:** 2025-04-14

**Authors:** Yi Liu, Cui Wang, Mingxia Li, Yunzhou Yang, Huiying Wang, Shufang Chen, Daqian He

**Affiliations:** 1Institute of Animal Husbandry and Veterinary Science, Shanghai Academy of Agricultural Sciences, Shanghai 201106, China; liuyi20031194@163.com (Y.L.); cuiwang518@saas.sh.cn (C.W.); kobebryant198687@163.com (Y.Y.); yjshywang@sina.com (H.W.); 2Academy of Life Sciences and Technology, Tongji University, Shanghai 200092, China; 2110839@tongji.edu.cn; 3NingBo Academy of Agricultural Sciences, Ningbo 315040, China

**Keywords:** goose, skeletal muscle satellite cells, transcriptomic, metabolomic, gene expression, differentiation, PPAR signaling pathway

## Abstract

Skeletal muscle satellite cells (SMSCs) are quiescent stem cells located in skeletal muscle tissue and function as the primary reservoir of myogenic progenitors for muscle growth and regeneration. However, the molecular and metabolic mechanisms governing their differentiation in geese remain largely unexplored. This study comprehensively examined the morphological, transcriptional, and metabolic dynamics of goose SMSCs across three critical differentiation stages: the quiescent stage (DD0), the differentiation stage (DD4), and the late differentiation stage (DD6). By integrating transcriptomic and metabolomic analyses, stage-specific molecular signatures and regulatory networks involved in SMSC differentiation were identified. Principal component analysis revealed distinct clustering patterns in gene expression and metabolite profiles across these stages, highlighting dynamic shifts in lipid metabolism and myogenesis. The PPAR signaling pathway emerged as a key regulator, with crucial genes such as *PPARG*, *IGF1*, *ACSL5*, *FABP5*, and *PLIN1* exhibiting differentiation-dependent expression patterns. Notably, *PPARG* and *IGF1* displayed negative correlations with adenosine and L-carnitine levels, suggesting their role in metabolic reprogramming during myotube formation. Additionally, *MYOM2* and *MYBPC1* exhibited stage-specific regulation and positively correlated with 2,3-dimethoxyphenylamine. This study provides a foundational framework for understanding muscle development and regeneration, offering valuable insights for both agricultural and biomedical research.

## 1. Introduction

Skeletal muscle, a vital organ for body movement and energy metabolism, comprises approximately 30–40% of an animal’s total body weight [1]. In meat-producing animals, skeletal muscle development is closely linked to meat yield and quality, both of which are economically significant traits in the livestock and poultry industries [2].

Skeletal muscle satellite cells (SMSCs) are a type of myoblast and the primary stem cells of skeletal muscle. They are located between the basal lamina and sarcolemma of muscle fibers and possess the potential for differentiation and proliferation. SMSCs were first discovered on the surface of frog tibialis anterior muscle cells [3] and were later isolated from adult rats, humans, cattle, sheep, chickens, pigs, ducks, geese, and pigeons [4,5,6,7,8,9,10,11]. Studies have demonstrated that the number of muscle fibers does not increase significantly with age following birth [3,12]. The growth in muscle volume and weight is primarily attributed to the proliferation and differentiation of muscle stem cells, ultimately achieving maturation through their fusion into muscle fibers [13]. Consequently, SMSCs play a crucial role in skeletal muscle growth, regeneration, and maintenance after birth [14]. Therefore, owing to their differentiation potential, satellite cells have emerged as a crucial cell model for the study of adult skeletal muscle and have been extensively utilized in investigating the mechanisms underlying muscle tissue formation and development [15,16,17,18]. However, existing studies on the isolation and identification of satellite cells have primarily concentrated on these fundamental processes. In-depth investigations into the regulatory mechanisms governing the proliferation and differentiation of satellite cells remain limited in the current literature.

In poultry studies, muscle satellite cells are usually obtained through isolating skeletal muscle tissue from early embryos. In geese, myosatellite cells are predominantly isolated from embryos ranging in age from 15 to 20 days [11]. Currently, the prevailing method for identifying satellite cells involves labeling specific genes through immunofluorescence techniques. Satellite cells remain in a quiescent state and exhibit high expression levels of the marker protein Pax7 [12,19]. Upon activation, these cells begin to express myogenic regulatory factors such as MyoD and MyoG, progressively differentiating into myoblasts and myocytes [20,21]. Therefore, in this study, the leg muscle of a 16-day-old embryo was utilized as the material for isolating and purifying myosatellite cells. Through morphological observation and immunofluorescence labeling of specific genes, myosatellite cells were categorized into three critical stages during their proliferation and differentiation process, which were designated as the quiescent stage (DD0), the differentiation stage (DD4), and the late differentiation stage (DD6). Through comprehensive morphological analysis, ultrastructural examination, gene expression profiling, and metabolomics investigations, the dynamic changes and regulatory networks of goose SMSCs during these three critical periods were systematically elucidated. This research provides a robust scientific foundation for further exploration of the molecular mechanisms and metabolic regulation underlying SMSC differentiation.

## 2. Results

### 2.1. Morphological Observation, Identification, and Differentiation of Goose SMSCs

The isolated SMSCs were cultured in a growth medium and began adhering after 12 h. By 24 h, the cells started to grow in a dispersed manner, with approximately 90% adhering to the surface. At 36 h, all cells had fully adhered and gradually extended into an elongated shape. By 48 h (DD0), cell confluence reached approximately 90% (Figure 1). Immunofluorescence staining confirmed Pax7 expression in the isolated cells, verifying that they were goose myoblasts, in which the positive rate of Pax7 was 69.23% (Figure 1). Additionally, the high purity of the isolated SMSCs ensured a reliable cell source for subsequent experiments.

When cell confluence reached 80–90%, SMSCs were induced to differentiate in vitro. After 12 h, the cells began to elongate gradually. By 48 h (DD4), small muscle tubes had formed and started to grow. At 72 h (DD6), the cells continued to proliferate and fuse, forming dense, mature muscle tubes, which eventually began to detach (Figure 1). Immunofluorescence staining detected high myoglobin expression in the differentiated SMSCs; myoglobin-positive cells were detected at 89.34% and 94.39% on DD4 and DD6, respectively (Figure 1). These results demonstrate that the isolated SMSCs possess strong differentiation potential.

### 2.2. Cellular Structure Changes and Apoptosis Following the Induced Differentiation of Goose SMSCs

Ultrastructural examination via electron microscopy revealed that the densely packed cells on DD0 exhibited intact mitochondrial and endoplasmic reticulum structures. (Figure 2A). By DD4, a decline in cell population was observed, accompanied by progressive atrophy and shrinkage of mitochondria and the endoplasmic reticulum. Additionally, a subset of cells exhibited apoptotic characteristics (Figure 2A). These findings suggest a potential suppression of cellular energy metabolism and protein synthesis functions. As differentiation advanced to DD6, a dramatic reduction in cell population occurred, characterized by the dissolution of intracellular mitochondria and endoplasmic reticulum, the nascent formation of myofibrillar nodes, and widespread apoptotic detachment of cells (Figure 2A).

Supporting these observations, flow cytometric analysis revealed a significant increase in the apoptotic cell ratio on DD4 and DD6 (Figure 2B,C), indicating that apoptosis accompanies satellite cell differentiation. This process may represent a cellular response to differentiation signals or microenvironmental changes. Furthermore, compared to DD0, mechanical cell mortality rates were markedly higher on DD4 and DD6 (Figure 2B,D), suggesting that morphological remodeling and mechanical stress during differentiation may collectively contribute to cell death. These findings imply that dysregulated energy metabolism or differentiation failure may be primary drivers of programmed cell death in this context.

Cell cycle analysis showed no significant differences in cell cycle distribution between DD4 and DD6. However, most cells were arrested in the G0–G1 phase, with only a small fraction progressing to the S phase (Figure 2E,F). This indicates a pronounced suppression of proliferative capacity during differentiation, as the majority of cells exited the proliferative phase and entered either a differentiated or apoptotic state. This phenomenon may be attributed to the dominant influence of differentiation signals, which likely inhibit proliferation to allocate resources for differentiation.

Collectively, these analyses revealed significant morphological and functional changes in SMSCs during induced differentiation, highlighting the dynamic evolution of cellular states throughout the process. Based on these findings, three key stages—DD0, DD4, and DD6—were selected as cell models for studying differentiation in vitro.

### 2.3. Analysis of DEGs in SMSCs in Different Periods

To elucidate the transcriptomic characteristics of SMSCs across different developmental stages, tissue samples were systematically collected at three time points: DD0 (n = 6), DD4 (n = 6), and DD6 (n = 6). A total of 18 samples underwent mRNA-Seq analysis, and all raw data were deposited in the SRA database (accession number: PRJNA1223016).

Principal component analysis (PCA) revealed distinct clustering of SMSC transcriptomes at each stage, indicating significant differences in gene expression patterns across developmental periods (Figure 3A). Differential gene expression analysis, based on mRNA-Seq data, identified variations in gene expression between time points through pairwise comparisons, using a significance threshold of |log_2_ fold change| > 1 and *p*-adjust < 0.05.

A Venn diagram illustrated that 652 genes were differentially expressed in both the DD0 vs. DD4 and DD0 vs. DD6 comparisons. Additionally, 1276 differentially expressed genes (DEGs) overlapped between the DD0 vs. DD4 and DD4 vs. DD6 comparisons, while 618 genes were shared between the DD0 vs. DD6 and DD4 vs. DD6 comparisons. Notably, 280 genes were common across all three comparisons (Figure 3B).

The clustering heatmap demonstrated a clear and consistent separation of SMSCs by developmental stage, with each tissue type forming a distinct cluster (Figure 3C). Enrichment analysis further revealed that pathways related to lipid metabolism, such as the PPAR signaling pathway, were significantly enriched in the G-C51 cluster (Figure 3C).

### 2.4. Identification of Key Regulatory Genes Associated with the PPAR Signaling Pathway

To further explore the biological significance of DEGs, Kyoto Encyclopedia of Genes and Genomes (KEGG) enrichment analyses were conducted. The results revealed that the most significantly enriched pathways in the comparison between DD0 and DD4 were neuroactive ligand–receptor interactions, cytokine–cytokine receptor interactions, and cell adhesion molecules (Figure 4A). These pathways also exhibited the highest levels of enrichment in the comparison between DD4 and DD6 (Figure 4B). Additionally, the PPAR signaling pathway, which is associated with lipid metabolism, was consistently enriched across all comparisons, suggesting that SMSCs may regulate lipid metabolism through PPAR signaling.

To further examine gene expression changes within the PPAR signaling pathway at different SMSC developmental stages, eight key genes closely linked to this pathway were selected for analysis. A differential gene volcano plot analysis revealed that, compared to SMSCs at the DD0 stage, the expression levels of peroxisome proliferative activated receptor gamma (*PPARG*), glucokinase (*GK*), fatty acid binding protein 5 (*FABP5*), acyl-CoA synthetase long chain family member 5 (*ACSL5*), DEP domain containing MTOR interacting protein (*DEPTOR*), insulin-like growth factor 1 (*IGF1*), and perilipin 1 (*PLIN1*) were significantly upregulated at DD4, whereas elongase of very long chain fatty acids 6 (*ELOVL6*) expression was markedly downregulated (Figure 4C). Conversely, when comparing DD4 to DD6, the expression of *PPARG*, *GK*, *FABP5*, *ACSL5*, *DEPTOR*, *IGF1*, and *PLIN1* was significantly downregulated, while *ELOVL6* expression was significantly upregulated (Figure 4D).

Interestingly, this study also identified myomesin-2 (*MYOM2*) and myosin binding Protein C1 (*MYBPC1*) as genes closely associated with SMSC differentiation. Notably, *MYOM2* expression was significantly upregulated at DD4 compared to its undifferentiated state at DD0. However, during the apoptotic phase (DD6), *MYOM2* expression was markedly downregulated, whereas *MYBPC1* expression remained relatively stable.

To validate the differential expression of these genes identified through transcriptome sequencing, quantitative real-time PCR (qRT-PCR) was performed to assess their expression levels across different developmental stages. The qRT-PCR results showed a strong correlation with the transcriptome analysis data (Figure 5). Specifically, at DD4, *PPARG*, *GK*, *FABP5*, *ACSL5*, *IGF1*, and *PLIN1* exhibited significantly elevated expression levels, whereas at DD6, their expression returned to levels comparable to those observed at DD0. Additionally, *DEPTOR* expression displayed a slight increase at DD6 relative to DD0. Notably, *ELOVL6* exhibited a distinct expression pattern, reaching its lowest level at DD4 and peaking at DD6. Furthermore, *MYOM2* and *MYBPC1* expression levels were significantly elevated at DD4 compared to DD0 but were markedly downregulated as SMSCs underwent apoptosis, aligning with the transcriptome analysis findings (Figure 5).

### 2.5. Metabolome Analysis and Identification of Key Metabolite in SMSCs in Different Periods

To systematically examine changes in the metabolite profiles of SMSCs across different developmental stages, comprehensive metabolomic sequencing was conducted on samples from three distinct stages. A total of 18 samples underwent metabolome analysis, and all raw data were deposited in the NGDC database (accession number: OMIX009001). PCA revealed well-defined clustering patterns, forming three distinct groups that reflect significant metabolic differences between stages (Figure 6A). The minimal variation among quality control (QC) samples further confirmed the reliability of our metabolomic data.

To gain deeper insight into the classification and functional properties of the identified metabolites, detailed classification and annotation were performed. The results showed that the most abundant metabolite categories were organic acids and derivatives (30.9%), organic heterocyclic compounds (18.6%), and lipids and lipid-like molecules (16.4%) (Figure 6B).

Furthermore, functional enrichment analysis of differentially abundant metabolites identified adenosine monophosphate (AMP) and L-carnitine as key components in lipid metabolism, while 2,3-methylenedioxyamphetamine was strongly associated with SMSC differentiation. Correlation analysis further revealed significant associations between these three metabolites and several others (Figure 6C). Figure 6D–F illustrate the relative abundance changes in AMP, L-carnitine, and 2,3-methylenedioxyamphetamine across developmental stages. Notably, both AMP and L-carnitine followed similar trends, reaching their lowest levels at the DD4 stage (*p* < 0.05) and peaking at the DD0 stage (Figure 6D,E).

### 2.6. Analysis of the Correlations Between Transcriptome and Metabolome

Differentially expressed metabolites (VIP > 1, *p* < 0.05) and genes (*p* < 0.05) were mapped to the KEGG pathway database. A correlation analysis (correlation coefficient > 0.8) was then performed to identify associations between genes and metabolites within shared pathways, resulting in a correlation network diagram. Figure 7 illustrates the relationships between the relative abundance changes in adenosine, L-carnitine, and 2,3-dimethoxyphenylamine and their corresponding differentially expressed genes.

In the comparison between DD0 and DD4, the DEGs *IGF1* and *PPARG*, associated with the PPAR signaling pathway (highlighted in green boxes), exhibited negative correlations with adenosine (Figure 7A). Similarly, *ACSL5*, *PLIN1*, and *PPARG*, also linked to the PPAR signaling pathway, showed negative correlations with L-carnitine (Figure 7B). During muscle satellite cell differentiation, *MYBPC1* and *MYOM2* displayed positive correlations with 2,3-dimethoxyphenylamine (Figure 7C).

In the DD4-DD6 comparison, *IGF1* and *PPARG* (highlighted in green boxes) again demonstrated negative correlations with adenosine (Figure 7D), while *ACSL5*, *PLIN1*, and *PPARG* maintained negative correlations with L-carnitine (Figure 7E), consistent with the DD0-DD4 findings. However, in this later stage, *MYBPC1* and *MYOM2* no longer exhibited significant correlations with 2,3-dimethoxyphenylamine during muscle satellite cell differentiation.

### 2.7. Validation of DEGs by qPCR

To verify the reliability of the sequencing data, 10 DEGs were randomly selected for qPCR verification. The results demonstrated that, in comparison to DD0, the expression levels of *DEPTOR*, *PPARG*, *FABP5*, *GK*, *MYOMC2*, and *PLIN1* genes were significantly upregulated in DD4, with an increase ranging approximately from five to seven times (Figure 5B–E,H,I). The upregulation of *ACSL5*, *IGF1*, and *MYBPC1* genes was markedly higher, reaching approximately 17 to 21 times (Figure 5A,F,G). Furthermore, the expression of the *ELOVL6* gene was significantly downregulated in DD4 (Figure 5J). In DD6, the expression level of the *MYBPC1* gene remained elevated, while the expression levels of the aforementioned genes were restored to levels comparable to those observed in DD0. The quantitative results of qPCR were highly consistent with the RNA-seq data. Compared to DD0, in addition to the significantly downregulated expression of the *ELOVL6* gene, the RNA-seq results (Figure 4C) demonstrated that several other genes, including *DEPTOR*, *PPARG*, *GK*, *FABP5*, *ACSL5*, *IGF1*, *MYOM2*, *MYBPC1*, and *PLIN1*, exhibited significantly upregulated expression levels at DD4. Conversely, when comparing DD6 to DD4, the expression levels of *PPARG*, *GK*, *FABP5*, *ACSL5*, *DEPTOR*, *IGF1*, and *PLIN1* were significantly downregulated, while *ELOVL6* expression was significantly upregulated (Figure 4D).

## 3. Discussion

This study conducted a comprehensive investigation into the morphological changes, transcriptional profiles, and metabolite dynamics of goose SMSCs across three key differentiation stages: DD0 (quiescent stage), DD4 (differentiation stage), and DD6 (late differentiation stage). This was achieved through morphological analysis, ultrastructural examination, gene expression profiling, and metabolomic analysis. By integrating mRNA sequencing and metabolomics data, key regulatory genes and metabolites associated with lipid metabolism and myogenic differentiation were identified. These findings provide a crucial foundation for understanding the molecular and metabolic mechanisms governing SMSC differentiation.

SMSCs are adult stem cells located between the basal lamina and the sarcolemma of skeletal muscle fibers. They are named for their distinctive positioning, which resembles a satellite orbiting muscle fibers [3,22]. Derived from mesodermal progenitor cells during embryogenesis, satellite cells play a critical role in skeletal muscle growth, development, and repair [23]. Their development during the embryonic period involves multiple stages, from early differentiation and proliferation to myotube formation and muscle fiber maturation [24]. This process is tightly regulated by various hormones and transcription factors [25].

Under normal physiological conditions, satellite cells remain in a quiescent state, expressing high levels of *Pax7* to maintain their stem cell properties [26]. However, upon skeletal muscle injury or stimulation by growth factors, they become activated and begin expressing myogenic regulatory factors such as *MYOD* and *MYOG* [27]. Subsequently, they differentiate into myoblasts and myocytes, which then fuse to form myotubes. During the induced differentiation of goose embryo satellite cells, significant morphological changes were observed in some cells at DD4 and DD6, exhibiting preliminary characteristics of apoptosis. Furthermore, as the culture duration extended, the majority of the cells gradually detached and lost viability. Subsequent flow cytometry analysis confirmed the presence of apoptosis at both DD4 and DD6 during differentiation induction; however, the difference between the two groups was not significant (*p* > 0.05). This outcome may be attributable to the removal of floating, deactivated cells during the experimental procedure. Mammalian satellite cells are differentiated and cultured in vitro for over 10 days [28], and this variation may be influenced by factors such as donor origin, culture conditions, and temperature [29,30]. Unfortunately, apoptosis was not employed as the primary observation measure in this study, nor were the transcriptional changes in apoptotic cells thoroughly investigated. In addition, given the paucity of research on the differentiation and apoptosis of goose satellite cells, it remains challenging to definitively ascertain the precise causes of these differences. Therefore, in the future, we should aim to conduct a comprehensive and in-depth investigation into the dynamic changes in morphological characteristics and transcriptional profiles during apoptosis, as well as further elucidate the underlying mechanisms of this process.

PCA of transcriptomic and metabolomic data revealed distinct clustering of SMSCs across the three developmental stages: DD0, DD4, and DD6 (Figure 3A and Figure 5A). These findings indicate substantial changes in gene expression and metabolic profiles at each stage, highlighting the presence of distinct molecular programs driving SMSC differentiation. Notably, the PPAR signaling pathway emerged as one of the most significantly enriched pathways, with key genes exhibiting dynamic expression patterns (Figure 3C and Figure 4A,B). This underscores the critical role of PPAR signaling in SMSC differentiation, consistent with previous studies demonstrating its pivotal function in muscle development and energy homeostasis [31,32].

Peroxisome proliferator-activated receptors (PPARs), particularly *PPARG*, a key member of the PPAR family, have been shown both in vivo and in vitro to regulate astrocyte proliferation during muscle regeneration. *PPARG* influences adipogenesis and myogenesis by modulating the expression of genes involved in lipid metabolism and mitochondrial function [33,34,35]. At DD4, key genes in the PPAR signaling pathway, including *PPARG*, *GK*, *FABP5*, *ACSL5*, *IGF1*, and *PLIN1*, were significantly upregulated, suggesting their role in promoting lipid utilization during SMSC differentiation and active muscle fiber formation. Conversely, their downregulation at DD6 indicates a potential shift away from lipid metabolism as differentiation progresses, favoring other processes such as apoptosis and aging. These findings align with the established roles of *PPARG* and *IGF1* in regulating muscle differentiation and lipid metabolism.

In SMSCs, *PPARG* serves as a major regulator of adipocyte differentiation [36] and contributes to muscle fat infiltration during regeneration [37]. Muscle-specific overexpression of *PPARG* promotes fat deposition by activating adipocyte differentiation regulators and enhancing the expression of *LPL*, *FABP4*, and *PLIN1* [38]. Conversely, downregulation of *PPARG* reduces the expression of fat metabolism-associated genes such as *ACSL*, *PLIN2*, and *FABP4*, thereby limiting muscle lipid deposition [39]. Consistent with our findings, the expression patterns of fat metabolism-related genes *FABP5*, *ACSL5*, and *PLIN1* mirrored those of *PPARG* across the three developmental stages, showing significant upregulation post-differentiation followed by downregulation in later stages. These observations underscore the crucial role of *PPARG* in SMSC differentiation and its regulatory function in lipid metabolism alongside other lipid metabolism-related genes.

Furthermore, in SMSCs, *IGF-1* promotes proliferation and differentiation by regulating the cell cycle [40,41]. In differentiated rat myoblasts, elevated *IGF-1* expression significantly enhanced myotube hypertrophy and myoprotein synthesis [42]. Additionally, *IGF-1* facilitates skeletal muscle regeneration and increases protein synthesis by activating the PI3K/AKT/MTOR and PI3K/AKT/GSK3β signaling pathways [17]. This study observed a significant upregulation of *IGF-1* expression during the differentiation phase of SMSCs, consistent with its role in promoting their activation. Integrated transcriptomic and metabolomic analyses revealed a negative correlation between *IGF-1* and *PPARG* with adenosine (Figure 4D), while *ACSL5*, *PLIN1*, and *PPARG* also exhibited negative correlations with L-carnitine. These findings suggest that adenosine and L-carnitine play critical roles in energy metabolism and the functional maintenance of skeletal muscle. SMSC differentiation is accompanied by substantial metabolic adjustments. At the onset of differentiation, AMP levels transiently increase due to heightened energy demands (increased ATP consumption), leading to activation of the AMPK signaling pathway [43,44]. As differentiation progresses, AMPK activity diminishes (reflecting reduced AMP levels), and the mTORC1 pathway is reactivated to support protein synthesis and mature muscle fiber formation [43]. This aligns with our findings, where AMP levels were markedly reduced during the DD0 stage, coinciding with the formation of myotubes and muscle fibers. Additionally, SMSC differentiation is associated with metabolic reprogramming, an increased NAD^+^/NADH ratio, enhanced mitochondrial oxidative metabolism, and decreased intracellular L-carnitine levels [45,46]. In this study, a significant reduction in L-carnitine levels was observed during the DD0 stage, consistent with previous research.

Notably, *MYOM2* and *MYBPC1* play critical roles in SMSC differentiation. *MYOM2* encodes actin-binding protein-2 (also known as M-protein), which is primarily expressed in fast-twitch skeletal muscle fibers [47]. It contributes to the three-dimensional arrangement of actin filaments and helps maintain muscle fiber structure and function in chicken myoblasts [48], pig skeletal muscle [49], and fish skeletal muscle [50]. *MYBPC1* encodes myosin-binding protein C, a key regulator of filament organization that facilitates actin–myosin cross-bridge formation during skeletal muscle contraction [51]. Our results showed that *MYOM2* expression was upregulated at DD4 but significantly downregulated at DD6, consistent with the apoptotic phase of differentiation. In contrast, *MYBPC1*, a key structural protein in the sarcomere, remained relatively stable throughout differentiation, indicating its persistent role in maintaining muscle fiber integrity. Integrated transcriptomic and metabolomic analyses further revealed a positive correlation between *MYBPC1* and *MYOM2* with 2,3-dimethoxyaniline, suggesting that this metabolite may promote myosinocyte differentiation. However, further investigation is needed to elucidate its precise mechanism and confirm its function in muscle biology. Collectively, these findings provide valuable insights into the molecular and metabolic mechanisms underlying SMSC differentiation, particularly highlighting the role of the PPAR signaling pathway. This study establishes a crucial theoretical foundation for future research on muscle development and regeneration.

## 4. Materials and Methods

### 4.1. Validation Isolation, Culture, and Differentiation of Goose SMSCs

Gander fertilized eggs (16 days post-incubation) were subjected to molecular sex identification using *CHD* gene primers (Table 1). After identification, the embryos were disinfected with ethanol before isolating SMSCs. The leg muscle of the goose embryo was dissected, and blood vessels, fat, and connective tissue were carefully removed. The muscle tissues were then minced into a paste and digested with 2 mg/mL Dispase II (Roche, Basel, Switzerland) and 4 mg/mL Collagenase II (Gibco, Grand Island, NY, USA) in high-glucose DMEM (Corning, Grand Island, NY, USA) at 37 °C for 50 min. To terminate digestion, high-glucose DMEM containing 10% FBS (Lonsera, Uruguay, South America) was added. The suspension was filtered through a 70 μm mesh sieve and centrifuged at 350× *g* for 8 min at room temperature. Red blood cells were removed using ACK lysis buffer (Gibco, Grand Island, NY, USA).

The remaining cells were resuspended in DMEM/F12 medium (Gibco, 11320033) supplemented with 10% FBS, 1% PS (Gibco, Grand Island, NY, USA), and 5 ng/mL bFGF (RD, St. Paul, MN, USA) and cultured in a Thermo Forma incubator (Thermo Fisher, Waltham, MA, USA) at 37 °C with 5% CO_2_. After one hour, fibroblasts adhered to the bottom of the culture flask, while SMSCs remained in suspension. The supernatant was transferred to a new Petri dish, and this process was repeated twice to further enrich satellite cells and eliminate fibroblasts.

To induce differentiation, the culture medium was replaced with differentiation medium containing 2% horse serum (Hyclone, Logan, UT, USA) and 1% PS when the cell density reached 70–80%. During cell culture and differentiation, Petri dishes were pre-coated with a solution of matrix glue (Corning, Grand Island, NY, USA) and DMEM at a 1:24 ratio.

### 4.2. Immunofluorescence Staining

Immunofluorescence was performed to identify the isolated SMSCs. Briefly, cells grown in a 6-well plate were first washed three times with PBS and then fixed with 4% paraformaldehyde for 20 min. After another three washes with PBS, the cells were permeabilized with 0.25% Triton X-100 in PBS for 10 min and subsequently blocked with blocking solution (2% BSA + 0.05% Triton X-100 in PBS) for 60 min at room temperature. Next, the cells were incubated overnight at 4 °C with primary anti-Pax7 antibody (Abcam, Cambridge, UK) and anti-Myog (Servicebio, Wuhan, China). After washing, the cells were incubated with a fluorescent secondary antibody (1:2000 dilution, Thermo Fisher, Waltham, MA, USA) for 1 h at room temperature. Following another wash, the cells were stained with 1× DAPI in PBS (10 μg/mL) for 20 min in the dark. Finally, the samples were imaged using a fluorescence microscope (OLYMPUS, Tokyo, Japan), and the proportion of positive cells was quantified using the Aipathwell v2 software (Servicebio, Wuhan, China).

### 4.3. Cell Cycle and Apoptosis Detection

For the apoptosis experiment, cells were analyzed using the PE Annexin V Apoptosis Detection Kit (BD Pharmingen, San Jose, CA, USA). Cells were washed twice with cold PBS and resuspended in 1× Binding Buffer at a concentration of 1 × 10^6^ cells/mL. A 100 µL aliquot of the cell suspension (1 × 10^5^ cells) was transferred into a 5 mL culture tube. Subsequently, 5 µL PE Annexin V and 5 µL 7-AAD were added to the tube. The cells were gently vortexed to ensure thorough mixing and incubated for 15 min at room temperature in the dark. After incubation, 400 µL of 1× Binding Buffer was added to each tube. The samples were analyzed using the LSRFortessa^TM^ X-20 Cell Analyzer (BD Biosciences, Milpitas, CA, USA) for 1 h.

For BrdU labeling, BrdU (Sigma, San Francisco, CA, USA) was diluted in freshly pre-warmed growth medium to a final concentration of 0.03 mg/mL. The BrdU-containing medium was added to the cells, which were then incubated at 37 °C for 90 min. After incubation, 1 mL of 1× FOXP3 Fix/Perm buffer (BioLegend, San Diego, CA, USA) was added to the collected cells, gently vortexed, and incubated at room temperature in the dark for 20 min. The cells were centrifuged, and the supernatant was discarded.

The cell pellet was washed once with FACS buffer (by centrifugation at 250× *g* for 5 min), followed by the addition of 500 µL of 2M HCl. The mixture was incubated in the dark at room temperature for 30 min with thorough mixing. The cells were then washed twice with FACS buffer and once with 1× FOXP3 Perm buffer (BioLegend, San Diego, CA, USA). The pellet was resuspended in 1 mL of 1× FOXP3 Perm buffer and incubated at room temperature in the dark for 15 min. After centrifugation and removal of the supernatant, the pellet was resuspended in 100 µL of 1× FOXP3 Perm buffer.

Next, an appropriate amount of fluorochrome-conjugated BrdU antibody (BioLegend, San Diego, CA, USA) was added, and the cells were incubated at room temperature in the dark for 30 min. The cells were washed twice with FACS buffer, followed by the addition of DAPI (1 million cells + 1 µL DAPI in FACS buffer), and incubated at room temperature for 10 min. Finally, the cells were washed in FACS buffer and analyzed using the LSRFortessa^TM^ X-20 Cell Analyzer (BD Biosciences, Milpitas, CA, USA).

### 4.4. TEM Staining for Cells

The cell pellet was collected after centrifugation and resuspended in TEM fixative (Servicebio, Wuhan, China). After fixation, 0.1 M phosphate buffer (PB, pH 7.4) was added, and the suspension was washed in PB for 3 min. The cell pellet was then encapsulated in 1% agarose before solidification.

Samples were fixed in 1% OsO_4_ (Ted Pella Inc, Redding, CA, America) in 0.1 M PB for 2 h at room temperature, protected from light, and then rinsed three times with PB. Dehydration was carried out using a graded acetone series, followed by stepwise infiltration with EMBed 812 resin (SPI, McClellan Park, CA, USA). The samples were embedded in pure resin and cured overnight at 37 °C. Polymerization was completed at 60 °C for over 48 h.

Resin blocks were sectioned into ultrathin slices (60–80 nm) and mounted on copper grids. Sections were stained with uranyl acetate and lead citrate, air-dried, and examined under a transmission electron microscope (TEM, Tokyo, Japan). Images were captured as needed.

### 4.5. Total RNA Isolation and Transcriptome Sequencing

Total RNA was extracted from muscle satellite cell samples using Trizol reagent (Invitrogen Life Technologies, Waltham, MA, USA) following the manufacturer’s protocol. The extracted RNA was stored at −80 °C for subsequent transcriptomic analysis. The RNA quantity and concentration were assessed using a NanoDrop NC2000 spectrophotometer (Thermo Fisher Scientific, Waltham, MA, USA), while RNA integrity was verified by agarose gel electrophoresis.

High-quality total RNA (3 µg) was used to construct the cDNA library using the NEBNext Ultra II RNA Library Prep Kit for Illumina (New England Biolabs Inc., Ipswich, MA, USA), following the manufacturer’s instructions. Double-stranded cDNA was purified, end-repaired, and size-selected for fragments ranging from 400 to 500 bp using AMPure XP beads. PCR amplification was performed, and the resulting products were purified again using AMPure XP beads to obtain the final target library.

Library quality was assessed using an Agilent 2100 Bioanalyzer (Agilent, Santa Clara, CA, USA) with an Agilent High Sensitivity DNA Kit. The effective library concentration was determined by real-time quantitative PCR using the StepOnePlus Real-Time PCR System (Thermo Scientific, Waltham, MA, USA). High-quality libraries were sequenced on the NovaSeq 6000 platform (Illumina, San Diego, CA, USA) at Shanghai Personal Biotechnology Co., Ltd., Shanghai, China.

### 4.6. Transcriptome Analysis

To ensure high-quality sequencing results, fastp (v0.22.0) was used for raw data preprocessing, including the removal of 3′ end adapter sequences and reads with average quality scores below Q20. Preprocessed reads were aligned with the reference genome (GenBank No. GCF_002166845.1, available at NCBI) using HISAT2 (v2.1.0) [52]. Gene expression levels were quantified using HTSeq (v0.9.1) [53], and read counts were normalized via the FPKM method.

Differential expression analysis was performed using DESeq (v1.38.3) [54], with selection criteria of |log_2_FoldChange| > 1 and an adjusted *p*-value < 0.05. Bidirectional clustering of DEGs was conducted using the Pheatmap R package (v1.0.12) [55], grouping genes into clusters based on similar expression patterns. Enrichment maps were generated to visualize significantly enriched terms within each cluster.

To elucidate the biological functions of DEGs, Gene Ontology (GO) enrichment analysis was conducted using topGO (v2.50.0) [56], with significant enrichment defined as an adjusted *p*-value < 0.05. KEGG pathway enrichment analysis was performed using clusterProfiler (v4.6.0) [57], focusing on pathways with adjusted *p*-values < 0.05. Additionally, gene set enrichment analysis (GSEA) was conducted using GSEA software (v4.1.0) [58], and the resulting enrichment pathway map was generated.

### 4.7. Metabolite Extraction and Detection

For intracellular metabolite extraction, 50 mg of muscle satellite cells was weighed into a 2 mL centrifuge tube, followed by the addition of 200 µL pre-chilled water and two steel beads. The sample was homogenized at 55 Hz for 60 s, which was repeated once, then mixed with 800 µL methanol–acetonitrile (1:1, *v*/*v*) and ultrasonicated for 30 min at room temperature. After freezing at −20 °C for 30 min, it was centrifuged at 12,000 rpm (4 °C) for 10 min. Then, 800 µL supernatant was collected, vacuum-dried, and resuspended in 150 µL of 50% methanol containing 5 ppm 2-chlorophenylalanine. After vortexing (30 s) and centrifugation (12,000 rpm, 4 °C, 10 min), the supernatant was filtered (0.22 µm) and transferred to an HPLC vial. A pooled QC sample was prepared to assess instrument stability.

Metabolites were separated via Thermo Scientific Vanquish Flex UHPLC System (Thermo Fisher, Waltham, MA, USA) on an ACQUITY UPLC HSS T3 column (100 Å, 1.8 µm, 2.1 mm × 100 mm) at 40 °C. The flow rate was 0.4 mL/min with a 2 μL injection volume. The mobile phases consisted of 0.1% formic acid in water (A) and 0.1% formic acid in acetonitrile (B). The elution gradient was as follows: 0–1 min: 5% B; 1–7 min: linear increase to 95% B; 7–8 min: held at 95% B; 8.1–13 min: returned to 5% B.

Mass spectrometry was performed using a Thermo Scientific Orbitrap Exploris 120 (Thermo Fisher, Waltham, MA, USA) in both positive and negative ion modes Thermo Scientific Xcalibur 4.7 (Thermo Fisher, Waltham, MA, USA). The HESI source settings were a spray voltage, 3.5 kV (positive) and −3.0 kV (negative); sheath gas, 40 arb; auxiliary gas, 15 arb; capillary temperature, 325 °C; and auxiliary heater temperature, 300 °C.

The primary resolution was set to 60,000 with a scan range of 100–1000 *m*/*z*, AGC target set to default, and Max IT at 100 ms. The top four ions underwent HCD fragmentation (dynamic exclusion: 8 s), with secondary resolution at 15,000 and a collision energy of 30%. System stability was ensured by injecting 2–4 QC samples before analysis and one QC sample every 5–10 runs.

### 4.8. Metabolome Analysis

To ensure high-quality data acquisition, raw mass spectrometry data were systematically evaluated and filtered using Compound Discoverer^TM^ 3.3 (version 3.3.2.31, Thermo Fisher Scientific, Waltham, MA, USA) [59]. This process involved reducing background noise, eliminating low-quality peaks, and normalizing the total peak area. Metabolite identification was performed by referencing an in-house database and several authoritative online databases, including mzCloud (https://www.mzcloud.org/, accessed on 12 January 2025), LIPID MAPS (https://www.lipidmaps.org/, accessed on 2 January 2025), HMDB (https://hmdb.ca/, accessed on 2 January 2025), MoNA (https://mona.fiehnlab.ucdavis.edu/, accessed on 2 January 2025), and the NIST_2020_MSMS spectral library [60]. For accurate metabolite identification, the MS1 mass tolerance was set to 15 ppm, and the MS2 match factor threshold was 50.

Expression abundance density plots and violin plots were generated using ggplot2 (version 3.4.1). Differentially expressed metabolites were analyzed using a comprehensive suite of bioinformatics tools [61]. Clustering analysis of metabolite abundance values was conducted using the Pheatmap package (version 1.0.12) in R, generating detailed heatmaps. Overlaps of differentially expressed metabolites between groups were visualized using Venn diagrams (VennDiagram, version 1.7.3) and UpSet plots (UpSetR, version 1.4.0). Correlation analysis was performed using the corrplot package (version 4.0.3), while boxplots and violin plots were created with ggplot2 (version 3.4.1) to illustrate metabolite abundance changes across experimental groups.

To elucidate the functional roles of differential metabolites, KEGG pathway enrichment analysis was conducted using the clusterProfiler package (version 4.6.0), identifying significantly enriched metabolic pathways.

### 4.9. Analysis of the Correlations Between the Transcriptome and Metabolome

In the integrated metabolomics and transcriptomics analysis, the results were first consolidated from both datasets, and we conducted correlation and O2PLS analyses [62]. Differential metabolites and transcripts were then identified using Metscape 2 [63], and enzyme-related transcripts corresponding to these metabolites were extracted from the KEGG database (https://www.kegg.jp/dbget-bin/www_bfind?compound, accessed on 11 January 2025).

Pearson correlation coefficients between differential metabolites and transcripts were calculated using R’s cor function, enabling the construction of a correlation network. Differential metabolites and all transcripts were mapped to the KEGG pathway database to identify shared pathways, selecting genes with *p*-values < 0.05 and metabolites with VIP values > 1 and *p*-values < 0.05. Finally, metabolic pathways of these differential metabolites and their corresponding transcripts were visualized using the pathview package in R. A correlation network for shared pathways was generated, including only gene–metabolite correlations with coefficients > 0.8.

### 4.10. qRT-PCR Validation

To validate the RNA-seq data, 10 DEGs were selected for qRT-PCR analysis. Primers were designed using Oligo 5.0 software (Table 1). qPCR reactions were performed in 20 µL volumes using TB Green^®^ Premix Ex Taq^TM^ II (Takara, Qindao, China), with six technical replicates per sample. Relative gene expression levels were calculated using the 2^−ΔΔCt^ method, with β-actin as the reference gene. PCR reactions were run on a C1000 Touch thermal cycler (Bio-Rad) in a 384-well plate format under the following conditions: 95 °C for 10 min, followed by 40 cycles of 95 °C for 15 s and 60 °C for 60 s.

## 5. Conclusions

This study highlights the role of the PPAR signaling pathway, regulated by key genes such as *PPARG*, *IGF1*, *ACSL5*, *FABP5*, and *PLIN1*, in coordinating lipid metabolism and myogenic differentiation in goose SMSCs across quiescent (DD0), differentiated (DD4), and late (DD6) stages. Notably, *PPARG* and *IGF1* negatively correlate with adenosine and L-carnitine levels, suggesting their role in metabolic reprogramming during myotube formation. Additionally, *MYOM2* and *MYBPC1* exhibit stage-specific regulation and positively correlate with 2,3-dimethoxyphenylamine, implying its potential role in SMSC differentiation.

## Figures and Tables

**Figure 1 ijms-26-03710-f001:**
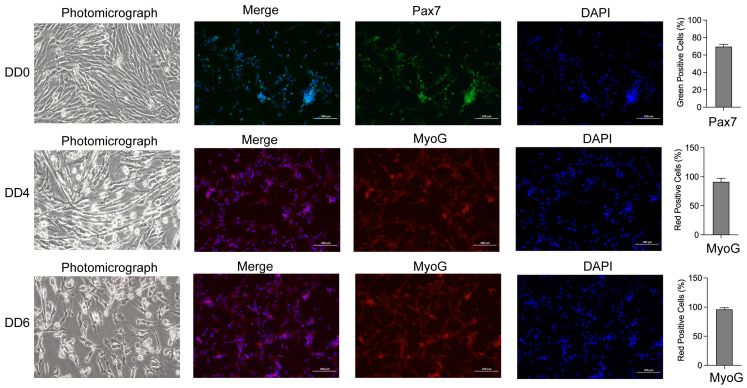
Morphological changes and immunofluorescence identification of goose SMSCs on DD0, DD4, and DD6 during induction of differentiation. Microscopic morphology displays that before induction (DD0), the cells are dense and grow well; after induction (DD4), the cells differentiate into dense and mature myotubes; by DD6, most of the myotubes have detached and died; before induction (DD0), *Pax7* is highly expressed in purified SMSCs; and after induction (DD4 and DD6), *MYOG* is highly expressed in differentiated SMSCs. Bar chart represents the proportions of Pax7-positive and MYOG-positive cells.

**Figure 2 ijms-26-03710-f002:**
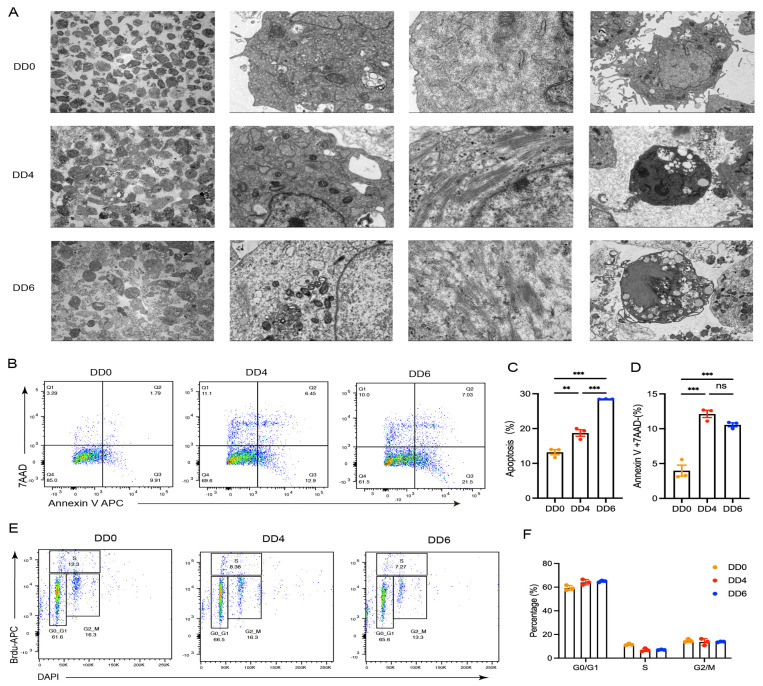
Ultrastructural observation by electron microscopy, cell cycle analysis, and apoptosis detection of goose SMSCs on DD0, DD4, and DD6. (**A**): Electron microscopy results show that SMSCs undergo significant morphological and structural changes during the induction of differentiation. On DD4 and DD6, the mitochondria and endoplasmic reticulum within the cells gradually shrink, while myofibrillar nodules begin to form, and the cells show a trend towards apoptosis. (**B**–**D**): Flow cytometry results and bar chart analysis indicate that the apoptotic trend of cells is significantly enhanced on DD4 and DD6 (**B**,**C**). In addition, compared with DD0, the mechanical cell death rate on DD4 and DD6 is significantly increased (**B**,**D**). (**E**,**F**): Cell cycle analysis results show that there is no significant change in the cell cycle on DD4 and DD6, but most cells are in the G0-G1 phase, with only a small number of cells entering the S phase. ns > 0.05, ** *p* < 0.01, *** *p* < 0.001.

**Figure 3 ijms-26-03710-f003:**
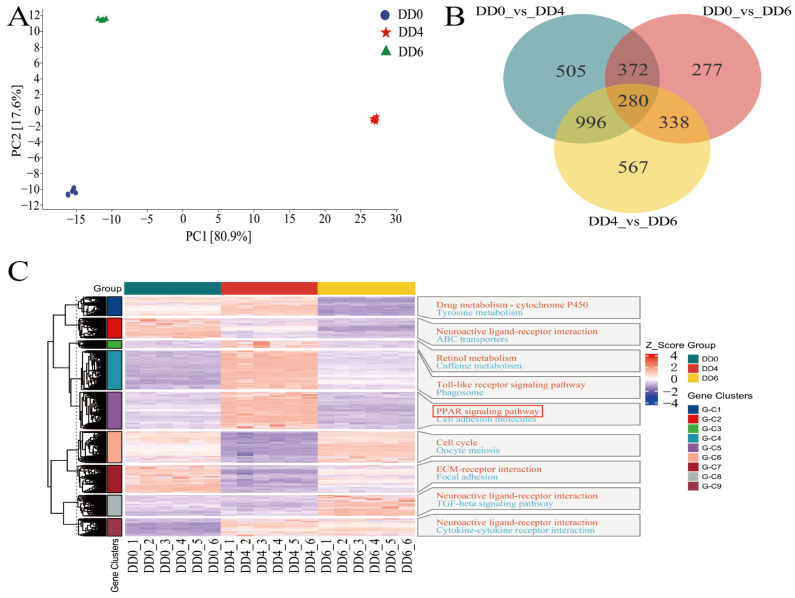
Transcriptome analysis of goose SMSCs during induction of differentiation. (**A**) Principal component analysis (PCA) of mRNA-Seq samples. (**B**) Venn diagram illustrating the overlap of differentially expressed genes (DEGs) between groups. (**C**) Hierarchical clustering analysis of DEGs and enriched pathways. The PPAR signaling pathway is marked with a red box. DD0: the quiescent stage; DD4: the differentiation stage; DD6: the late differentiation stage; n = 6.

**Figure 4 ijms-26-03710-f004:**
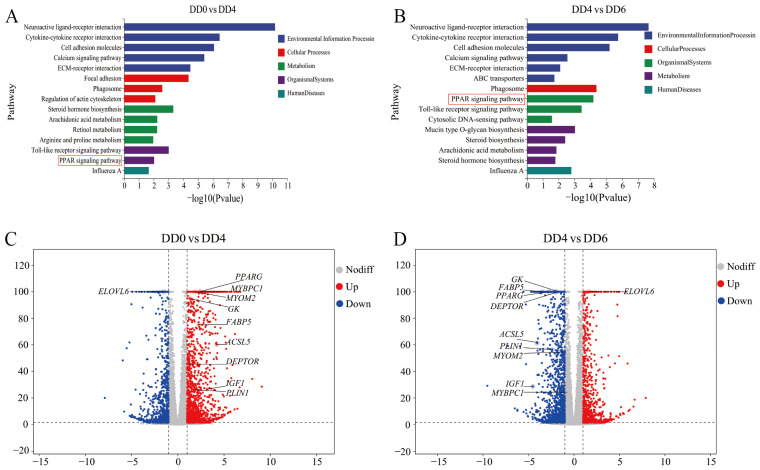
KEGG enrichment analysis and volcano plots of DEGs in goose SMSCs during induction of differentiation. (**A**) Enriched KEGG pathways comparing DD0 and DD4. The PPAR signaling pathway is marked with a red box. (**B**) Enriched KEGG pathways comparing DD4 and DD6. The PPAR signaling pathway is marked with a red box. (**C**) Volcano plot of DEGs comparing DD0 and DD4. (**D**) Volcano plot of DEGs comparing DD4 and DD6. DD0: the quiescent stage; DD4: the differentiation stage; DD6: the late differentiation stage; n = 6.

**Figure 5 ijms-26-03710-f005:**
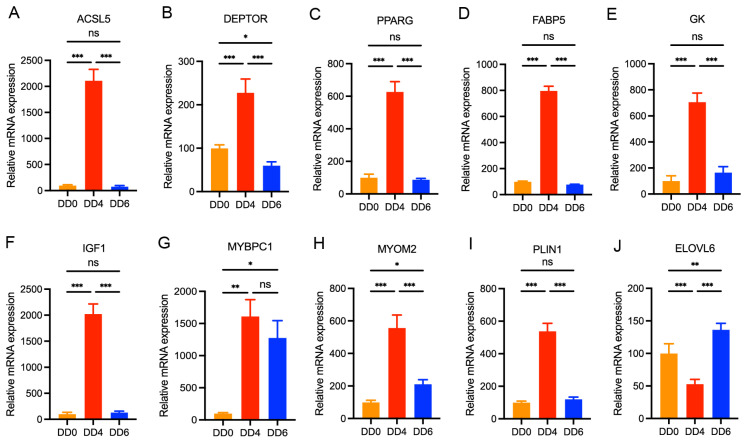
Relative quantitative values for ten selected DEG obtained from the RNA-seq data. (**A**–**J**) Relative mRNA expression levels of *ACSL5*, *DEPTOR*, *PPARG*, *FABP5*, *GK*, *IGF1*, *MYBPC1*, *MYOM2*, *PLIN1* and *ELOVL6* genes in goose SMSCs during induction of differentiation, respectively. ns > 0.05, * *p* < 0.05, ** *p* < 0.01, *** *p* < 0.001; DD0: the quiescent stage; DD4: the differentiation stage; DD6: the late differentiation stage; n = 6.

**Figure 6 ijms-26-03710-f006:**
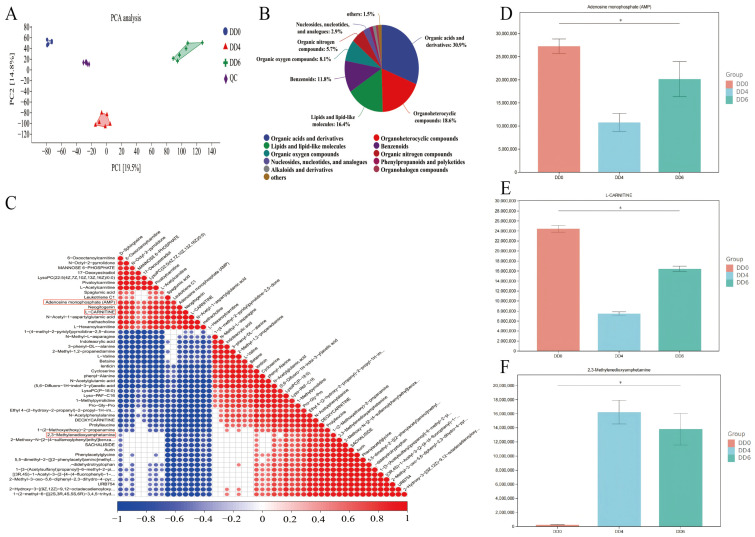
Metabolome analysis of goose SMSCs during induction of differentiation. (**A**) Principal component analysis (PCA) of metabolomic profiles. (**B**) Classification and identification of metabolites. (**C**) Correlation analysis of differentially abundant metabolites, illustrating the relationships between key metabolites. (**D**–**F**) Relative abundances of adenosine monophosphate (AMP), L-carnitine, and 2,3-Methylenedioxyamphetamine in goose SMSCs during induction of differentiation. DD0: the quiescent stage; DD4: the differentiation stage; DD6: the late differentiation stage; n = 6; * *p* < 0.05.

**Figure 7 ijms-26-03710-f007:**
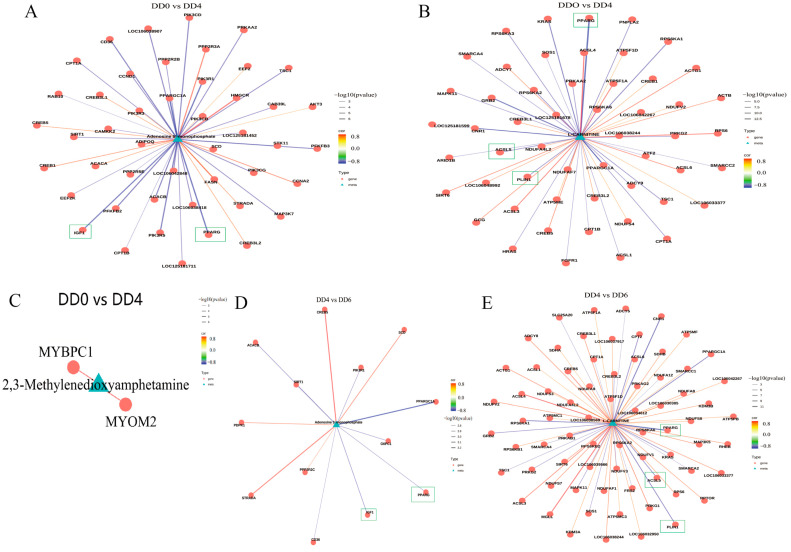
Transcriptome-metabolome correlation analysis. (**A**) DEGs associated with adenosine monophosphate (AMP) comparing DD0 and DD4. The *IGF1* and *PPARG* genes is marked with a green box. (**B**) DEGs associated with adenosine monophosphate (AMP) comparing DD4 and DD6. The *ACSL5*, *PLIN1* and *PPARG* genes is marked with a green box. (**C**) DEGs associated with 2,3-Methylenedioxyamphetamine comparing DD0 and DD4. (**D**) DEGs associated with L-carnitine comparing DD0 and DD4. The *IGF1* and *PPARG* genes is marked with a green box. (**E**) DEGs associated with L-carnitine comparing DD4 and DD6. The *ACSL5*, *PLIN1* and *PPARG* genes is marked with a green box. DD0: the quiescent stage; DD4: the differentiation stage; DD6: the late differentiation stage; n = 6.

**Table 1 ijms-26-03710-t001:** Sex determination and real-time PCR primer sequences.

Gene Name	Primer Sequences (5′–3′)	Annealing Temperature	Size of Target Fragments
*CHD*	F: TGCAGAAGCAATATTACAAGT	60 °C	466/326 bp
	R: AATTCATTATCATCTGGTGG		
*FABP5*	F: ACAATCACCGTAAAAACAGAAA	60 °C	186 bp
	R: AAGTTTCCGTGTTATTATGGTC		
*PPARG*	F: CAGGAGCAGAACAAAGAGGTAG	60 °C	185 bp
	R: GAAGCCAGGAGAGTATATATGA		
*DEPTOR*	F: CACGAGGAGAAGGTCATTAAGG	60 °C	128 bp
	R: TTAATTGCTGTCTCTCGGTCGG		
*ACSL5*	F: GGAAAGACCCCATGTGTGAAGA	60 °C	175 bp
	R: ACACAATGCAAAGATCTTCAGG		
*MYOM2*	F: AAGGATCCGGTTTGCCAGTGAG	60 °C	160 bp
	R: AGCTCGACTTATTCTTTCCTCA		
*MYBPC1*	F: CTGAAAAGGGCAAAGATGAAGA	60 °C	162 bp
	R: CGACGAATAAGGTGGATCTCTG		
*ELOVL6*	F: ACTGTACGCTGCCTTTATATTT	60 °C	115 bp
	R: AAGTATTCTGAAGACGGCAAGG		
*PLIN1*	F: GAGGGCTATGAGGCGACCAAGA	60 °C	165 bp
	R: CTTCTGATCTGCTTCCTCGTCC		
*GK*	F: AAGAAGGATGGGTGGAACAAGA	60 °C	195 bp
	R: ACACAATTGCGTTATAAAGAGG		
*IGF1*	F: TTCTTCTACCTTGGCCTGTGTT	60 °C	237 bp
	R: AGCACAGTACATCTCCAGCCTC		
*β-actin*	F: TCCGTGACATCAAGGAGAAG	60 °C	224 bp
	R: CATGATGGAGTTGAAGGTGG		

## Data Availability

The datasets utilized in this study are publicly accessible through online repositories. Specifically, the transcriptome data are deposited in the NCBI BioProject database under accession number PRJNA1223016 (https://www.ncbi.nlm.nih.gov/bioproject/PRJNA1223016, registered on 12 February 2025). The metabolome data are available in the National Genomics Data Center database with the identifier OMIX009001 (https://www.ebi.ac.uk/ebisearch/search?db=wgs_masters&query=OMIX009001, registered on 12 February 2025).

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
