# Peer review of "Unveiling Key Genes and Crucial Pathways in Goose Muscle Satellite Cell Biology Through Integrated Transcriptomic and Metabolomic Analyses"

_ijms, 2025, doi:10.3390/ijms26083710_

Round 1
Reviewer 1 Report
Comments and Suggestions for Authors
Liu et al. performed RNA sequencing (RNA-seq) and metabolomics analysis of embryonic goose SMSCs at day 0, 4 and 6 to identify signaling pathways and metabolic products associated with in vitro myogenic differentiation.
The introduction does not provide sufficient information on what is known (or not known) about myogenesis in goose. The last section describes the relevance of muscle growth and repair in context of meat quality. As this study is performed using embryonic goose SMSCs, more information should be provided to explain rationale.
Quantification is missing of figure 1 of the results section, e.g. % Pax7 and % Myog positive cells, myogenic fusion index. Accurate characterization of the cells is key for interpreting the RNAseq data correctly. DD6 is described as “late differentiation”, but it is unclear how this is determined. It appears that DD6 contains apoptotic cells rather than late differentiation myotubes. Figure 7 shows qPCR analysis of 10 DEGs, how does this confirm the high accuracy of the RNAseq analysis (line 264)? Also, brightfield images appear to be different fields of view compared to immunofluorescent images in figure 1, e.g. DAPI shows regions without any DAPI. Resolution of images in the manuscript is very poor and too small. Wasn't apoptosis pathway altered in RNAseq data?
Discussion doesn’t contain critical discussion of own data, e.g. the observed apoptosis at day 6 is not discussed. Based on data from myogenic differentiation of e.g. mouse myoblasts, myotubes can be cultured for >10 days. Similarities/differences of goose myogenisis compared to other animals needs to be discussed. Discussion doens't link back to relevance of study for meat quality.
Specific remarks:
#42 Rephrase, satellite cells are not a type of myoblasts
#62 “.. geese rely on skeletal muscle ….” unclear sentence
#63/64 reference missing
#78 Following in vitro proliferation, the term SMSCs is not appropriate, they are satellite cell derived myogenic precursors, or myoblasts.
#97 “… normal growth patterns ...” ?
Throughout the manuscript, a lot of gene symbols are used without mentioning full gene name first.
Gene expression data mentions “significantly up/downregulated”, addition of the fold change would be helpful.
Comments on the Quality of English LanguageLanguage is in general okay, some sentences could be improved.
Author Response
Comments 1: The introduction does not provide sufficient information on what is known (or not known) about myogenesis in goose. The last section describes the relevance of muscle growth and repair in context of meat quality. As this study is performed using embryonic goose SMSCs, more information should be provided to explain rationale.
Response 1: Thank you for pointing this out. We agree with this comment. Therefore, we have conducted a thorough revision of the introduction. This modification can be found in lines 51-103 of the revised manuscript. We hope the revision will meet your requirements and thank you again for your suggestions to improve our manuscript.
Comments 2: Quantification is missing of figure 1 of the results section, e.g. % Pax7 and % Myog positive cells, myogenic fusion index.
Response 2: Thank you for pointing this out. We agree with this comment. Therefore, We updated Figure 1 to incorporate the data for the percentage of Pax7-positive and MyoG-positive cells. However, we regret to note that the myogenic fusion index was not analyzed at that time. This alteration can be found in lines 111-112 and 118-123 of the revised manuscript.
Comments 3: Accurate characterization of the cells is key for interpreting the RNAseq data correctly. DD6 is described as “late differentiation”, but it is unclear how this is determined. It appears that DD6 contains apoptotic cells rather than late differentiation myotubes.
Response 3: Thank you for pointing this out. We categorized the proliferation and differentiation of myosatellite cells into three distinct stages using morphological observation and immunofluorescence labeling of specific markers. Notably, microscopic analysis revealed that after day 6 of differentiation culture, cell viability significantly declined on days 7 and 8, with muscle tubes beginning to detach. The figure below presents representative transmission electron microscopy (TEM) images from day 8 of differentiation and culture (not included in the manuscript), illustrating myotube detachment (A), a marked reduction in cell number (B), endoplasmic reticulum dilation (C), loss of myotube integrity (D), and apoptosis (E). Based on these observations, we conclude that by day 6 of differentiation culture, the cells had reached the terminal stage of differentiation and subsequently entered a phase of widespread apoptosis.
Comments 4: Figure 7 shows qPCR analysis of 10 DEGs, how does this confirm the high accuracy of the RNAseq analysis (line 264)?
Response 4: Thank you for pointing this out. We agree with this comment. Therefore, we have deleted the sentence “confirming the accuracy of the results of the RNA-seq”. This alteration can be found in lines 317-318 of the revised manuscript.
Comments 5: Also, brightfield images appear to be different fields of view compared to immunofluorescent images in figure 1, e.g. DAPI shows regions without any DAPI.
Response 5: Thank you for pointing this out. We agree with this comment. In Figure 1, the first column shows the morphological images, while columns 2, 3, and 4 present the immunofluorescence staining images. It should be noted that the morphological images and the immunofluorescence staining images correspond to different fields of view. Additionally, the brightfield images are not included in this figure.
Comments 6: Resolution of images in the manuscript is very poor and too small.
Response 6: Thank you for pointing this out. We agree with this comment. We sincerely apologize for the suboptimal resolution and reduced size of the images in the manuscript. In order to maintain data integrity and ensure high-quality presentation, we have uploaded the original high-resolution images separately to the Supplementary Materials section of the submission system.
Comments 7: Wasn't apoptosis pathway altered in RNAseq data?
Response 7: A minor degree of apoptosis was detected in DD6 samples. In transcriptomic analysis, when comparing DD6 with DD4, it was observed that certain genes were enriched in the apoptosis pathway; however, the extent of enrichment did not reach statistical significance (P = 0.46). Additionally, alterations in the expression levels of several apoptosis-related genes were detected. For instance, MAP2K2 and PARP3 exhibited upregulation, whereas CTS and PARP4 demonstrated downregulation. Given the absence of significant enrichment in the apoptosis pathway, the associated findings were excluded from the manuscript.
Comments 8: Discussion doesn’t contain critical discussion of own data, e.g. the observed apoptosis at day 6 is not discussed. Based on data from myogenic differentiation of e.g. mouse myoblasts, myotubes can be cultured for >10 days. Similarities/differences of goose myogenisis compared to other animals needs to be discussed. Discussion doens't link back to relevance of study for meat quality.
Response 8: Thank you for pointing this out. We agree with this comment. Therefore, we haved added “Notably, apoptosis was detected in a limited number of cells within DD6. Mammalian satellite cells are differentiated and cultured in vitro for over 10 days, and this variation may be influenced by factors such as donor origin, culture conditions, and temperature. However, given the paucity of research on the differentiation and apoptosis of goose satellite cells, it remains challenging to definitively ascertain the precise causes of these differences. Therefore, further investigations are warranted to elucidate the underlying mechanisms.” in discussion. We re-assessed the relevance of satellite cells to meat quality in the introduction and determined that this content should be excluded. Consequently, the analysis of the correlation between satellite cells and meat quality is no longer addressed in the discussion section. This alteration can be found in lines 337-343 of the revised manuscript.
Comments 9: #42 Rephrase, satellite cells are not a type of myoblasts
Response 9: Thank you for pointing this out. We agree with this comment. Therefore, the sentence “Skeletal muscle satellite cells (SMSCs) are a type of myoblast and the primary stem cells of skeletal muscle.” has been revised to "Skeletal muscle satellite cells (SMSCs) are quiescent stem cells located in skeletal muscle tissue and function as the primary reservoir of myogenic progenitors for muscle growth and regeneration." . This alteration can be found in lines 16-19 of the revised manuscript.
Comments 10:#62 “.. geese rely on skeletal muscle ….” unclear sentence
Response 10: Thank you for pointing this out. We agree with this comment. Therefore, we have conducted a thorough revision of the introduction. Also include this sentence.
Comments 11:#63/64 reference missing
Response 11: Thank you for pointing this out. We agree with this comment. Therefore, we have conducted a thorough revision of the introduction. Also include this sentence.
Comments 12:#78 Following in vitro proliferation, the term SMSCs is not appropriate, they are satellite cell derived myogenic precursors, or myoblasts.
Response 12: Thank you for pointing this out. We agree with this comment. Therefore, the term “SMSCs” has been revised to "myoblasts". This alteration can be found in lines 111 of the revised manuscript.
Comments 13:#97 “… normal growth patterns ...” ?
Response 13: Thank you for pointing this out. We agree with this comment. Therefore, the sentence “Ultrastructural examination via electron microscopy revealed that cell colonies exhibited normal growth patterns at DD0, with densely packed cells displaying intact mitochondrial and endoplasmic reticulum structures” has been revised to "Ultrastructural examination via electron microscopy revealed that the densely packed cells at DD0 exhibited intact mitochondrial and endoplasmic reticulum structures.” This alteration can be found in lines 132-136 of the revised manuscript.
Comments 14: Throughout the manuscript, a lot of gene symbols are used without mentioning full gene name first.
Response 14: Thank you for pointing this out. We agree with this comment. Therefore, we have added the full name of these genes when them were first used. This alteration can be found in lines 217-227 of the revised manuscript.
Comments 15: Gene expression data mentions “significantly up/downregulated”, addition of the fold change would be helpful.
Response 15: Thank you for pointing this out. We agree with this comment. Therefore, we have added the fold change of gene expression in result. This alteration can be found in lines 309-316 of the revised manuscript.

Reviewer 2 Report
Comments and Suggestions for Authors
This study provides a comprehensive examination of the morphological, transcriptional, and metabolic dynamics of goose satellite muscle stem cells (SMSCs) across three critical differentiation stages: the quiescent stage (DD0), the differentiation stage (DD4), and the late differentiation stage (DD6). By integrating transcriptomic and metabolomic analyses, the authors identified stage-specific molecular signatures and regulatory networks governing SMSC differentiation. The study presents original findings that provide a foundational framework for understanding muscle development and regeneration, offering valuable insights for both agricultural and biomedical research. The methods employed are appropriate, the research quality is commendable, and the results are well-supported. However, the authors should carefully review the manuscript to address several minor issues, including grammatical errors, and improve overall clarity. I recommend accepting the manuscript after minor revisions. Specific areas requiring improvement are outlined below:
- Keywords: Consider adding "differentiation" and "PPAR signaling pathway" to enhance keyword relevance.
- Introduction (Lines 61-70): The discussion of recent advancements in SMSC research is insufficient, and the limitations of previous studies are not clearly emphasized. Please expand on previous research and explicitly state the scientific question of the current study.
- The rationale for selecting DD0, DD4, and DD6 as key stages is unclear. Please provide a clear explanation in either the Introduction or Materials and Methods section to justify why these time points were chosen for subsequent analysis.
- Line 144: Remove the abbreviation "(DGE)" from "Differential gene expression (DGE) analysis" to maintain consistency and readability.
- Line 272: The phrase "across three key differentiation stages" requires clarification. Are these stages considered "key" based on previous literature, or is this define by the authors? Please provide supporting evidence or justification.
- Formatting of Statistical Notation: Throughout the manuscript, ensure that "p" in "adjusted p-value < 0.05" is italicized.
Author Response
Comments 1: This study provides a comprehensive examination of the morphological, transcriptional, and metabolic dynamics of goose satellite muscle stem cells (SMSCs) across three critical differentiation stages: the quiescent stage (DD0), the differentiation stage (DD4), and the late differentiation stage (DD6). By integrating transcriptomic and metabolomic analyses, the authors identified stage-specific molecular signatures and regulatory networks governing SMSC differentiation. The study presents original findings that provide a foundational framework for understanding muscle development and regeneration, offering valuable insights for both agricultural and biomedical research. The methods employed are appropriate, the research quality is commendable, and the results are well-supported. However, the authors should carefully review the manuscript to address several minor issues, including grammatical errors, and improve overall clarity. I recommend accepting the manuscript after minor revisions.
Response 1: We are truly grateful for your comprehensive evaluation of our manuscript and for sharing your insightful and constructive feedback with us. We also deeply appreciate your acknowledgment of our efforts. Based on your suggestions, we have carefully revised the manuscript and included a detailed description of the changes. Additionally, we have clearly highlighted all modifications using our tracking system to make it easier for you to review.
Comments 2: Keywords: Consider adding "differentiation" and "PPAR signaling pathway" to enhance keyword relevance.
Response 2: Thank you for pointing this out. We agree with this comment. Therefore, we have added "differentiation" and "PPAR signaling pathway" in keyword. This alteration can be found in line 38 of the revised manuscript.
Comments 3: Introduction (Lines 61-70): The discussion of recent advancements in SMSC research is insufficient, and the limitations of previous studies are not clearly emphasized. Please expand on previous research and explicitly state the scientific question of the current study.
Response 3: Thank you for pointing this out. We agree with this comment. Therefore, we have conducted a thorough revision of the introduction. This modification can be found in lines 51-103 of the revised manuscript. We hope the revision will meet your requirements and thank you again for your suggestions to improve our manuscript.
Comments 4: The rationale for selecting DD0, DD4, and DD6 as key stages is unclear. Please provide a clear explanation in either the Introduction or Materials and Methods section to justify why these time points were chosen for subsequent analysis.
Response 4: Thank you for pointing this out. We agree with this comment. We sincerely thank you for raising this important point. In the revised manuscript, we have added detailed justifications for the selection of DD0, DD4, and DD6 as critical time points in the Introduction sections. This alteration can be found in line 88-98 of the revised manuscript.
Comments 5: Line 144: Remove the abbreviation "(DGE)" from "Differential gene expression (DGE) analysis" to maintain consistency and readability.
Response 5: Thank you for pointing this out. We agree with this comment. Therefore, we have removed the abbreviation "(DGE)" from "Differential gene expression (DGE) analysis". This alteration can be found in line 184 of the revised manuscript.
Comments 6: Line 272: The phrase "across three key differentiation stages" requires clarification. Are these stages considered "key" based on previous literature, or is this define by the authors? Please provide supporting evidence or justification.
Response 6: Thank you for pointing this out. We categorized the proliferation and differentiation of myosatellite cells into three distinct stages using morphological observation and immunofluorescence labeling of specific markers. Notably, microscopic analysis revealed that after day 6 of differentiation culture, cell viability significantly declined on days 7 and 8, with muscle tubes beginning to detach. The figure below presents representative transmission electron microscopy (TEM) images from day 8 of differentiation and culture (not included in the manuscript), illustrating myotube detachment (A), a marked reduction in cell number (B), endoplasmic reticulum dilation (C), loss of myotube integrity (D), and apoptosis (E). Based on these observations, we conclude that by day 6 of differentiation culture, the cells had reached the terminal stage of differentiation and subsequently entered a phase of widespread apoptosis.
Comments 7: Formatting of Statistical Notation: Throughout the manuscript, ensure that "p" in "adjusted p-value < 0.05" is italicized.
Response 7: Thank you for pointing this out. We agree with this comment. Therefore, we have changed "p" in "adjusted p-value < 0.05" is italicized throughout the manuscript.

Round 2
Reviewer 1 Report
Comments and Suggestions for Authors
Liu et al. adapted some aspects of the manuscript, e.g. MyoG and Pax7 quatification, although details on analysis is missing (how many fields of view, how many cells, standard deviation). Discussion, a few sentences have been added on apoptosis, but these were quite general remarks, not thorough critical discussion of own data.
qPCR correlation with RNAseq line 266?
Author Response
Comments 1: Liu et al. adapted some aspects of the manuscript, e.g. MyoG and Pax7 quatification, although details on analysis is missing (how many fields of view, how many cells, standard deviation).
Response 1: We appreciate the reviewer's approval of our revised content. In response to the reviewer's suggestion, we have made further improvements to Figure 1 and updated the corresponding caption (revised version, lines 107-108). Furthermore, a detailed description of the positive cell assay methods has been added in Section 4.2 of the "Materials and Methods" section (revised edition, lines 452-453).
Comments 2: Discussion, a few sentences have been added on apoptosis, but these were quite general remarks, not thorough critical discussion of own data.
Response 2: Thank you for pointing this out. We agree with this comment. Therefore, we have further revised and improved the discussion section to better align with your requirements. This alteration can be found in lines 322-340 of the revised manuscript.
Comments 3: qPCR correlation with RNAseq line 266?
Response 3: Thank you for pointing this out. We agree with this comment. We corrected this error and changed the word "correlation" to "consistency" (293).

Round 3
Reviewer 1 Report
Comments and Suggestions for Authors
Consistency should be consistent. How can readers check if it is consistent? Where can they find this data?
Comments on the Quality of English Languagesee above
Author Response
Comments 1: Consistency should be consistent. How can readers check if it is consistent? Where can they find this data?
Response 1: We truly appreciate your valuable feedback and meticulous guidance on our manuscript. We fully agree with this comment. As you suggested, we have detailed the consistency between the qPCR results and the RNA-seq data, and clearly labeled the data sources cited. These revisions can be found in lines 285-299 of the revised manuscript. Finally, we would like to express our sincere gratitude for your patient review and professional support. Your input has been incredibly helpful to us.

Round 4
Reviewer 1 Report
Comments and Suggestions for Authors
No additional comments